# Sustainable Digital Marketing Model of Geoenergy Resources under Carbon Neutrality Target

**Yingge Zhang [1,*], Zhihu Xia [2], Yanni Li [2], Anmai Dai [2] and Jie Wang [3]**

1.  School of Public Administration, Xiangtan University, Xiangtan 411100, China
2.  School of Business, Xiangtan University, Xiangtan 411100, China
3.  School of Electrical and Automation, Shandong University of Science and Technology, Qingdao 266400, China
*   Correspondence: 202021000574@smail.xtu.edu.cn

**Abstract:** Geoenergy resources are a new type of clean energy. Carbon neutralization and carbon peaking require significant system reform in the field of energy supply. As a clean, low-carbon, stable and continuous non carbon-based energy, geothermal energy can provide an important guarantee for achieving this goal. Different from the direct way of obtaining energy, ground energy indirectly obtains heat energy from shallow soil and surface water. The vigorous development of geoenergy resources under China's carbon neutrality goal plays an important role in energy conservation and emission reduction. However, the current carbon trading market is not understood by the public. Therefore, this paper aims to analyze the impact of geoenergy resources on promoting sustainable digital marketing models. Every country around the world is working hard to meet its carbon neutrality goal. This paper analyzed the economic goal of carbon neutrality by analyzing the principle of the carbon trading market. For this reason, this paper designed a carbon trading price prediction algorithm based on the BP neural network (BPNN), which can predict prices in the carbon trading market in order to promote the accurate push of the digital marketing model and finally drive ground energy resources to promote a sustainable digital marketing model. The experimental results of this paper have proved that the price error rate of the BPNN carbon trading price prediction algorithm designed in this paper was within 10%, which was about 20% smaller than the traditional multiple regression analysis algorithm. This proved that the algorithm in this paper has a good performance and can provide accurate information to allow the digital marketing model to achieve sustainable digital marketing.

**Keywords:** carbon neutrality target; geoenergy resources; digital marketing model; sustainable development; price prediction algorithm

## 1. Introduction

With the rapid development of the Internet and information technology, digital marketing has become a major trend under the background of digitalization guided by national policies. Compared with traditional marketing methods, digital marketing is more intelligent and efficient in various dimensions such as brand building, user expansion, and precision marketing, because digital marketing can greatly speed up the exchange of digital information and enhance the overall competitiveness of enterprises. In this article, research is presented on sustainable digital marketing models based on the goal of carbon neutrality and a BPNN-based carbon trading price algorithm is designed to explain the origins of the carbon neutral market. At present, digital marketing is still a new concept, and many companies are focused on digital marketing methods, such as online membership, display advertising, content marketing, search engine marketing (SEM), Search Engine Optimization (SEO), Email-making, analytics and so on. However, there is no mature and systematic framework guidance for practical application and digital marketing program design. For the field of industrial products, there is even less research related to digital

marketing, which makes it difficult to break through the existing marketing bottlenecks in the actual development of industrial enterprises. This not only reflects the difference in the development of the industry, but also reflects signs of talent shortage and insufficient research. As everyone knows, China is moving towards the goal of carbon neutrality. With the rapid development of the industrial Internet and the era of 5G Internet of Everything, digital sales of industrial products have become a top priority. In the entire industry chain, enterprises that can take the lead in realizing digital transformation will gain more opportunities with the rapid development of the future. Digital marketing occupies a pivotal position in the carbon trading market, and it is also an excellent perspective to use digital marketing to promote the rapid development of carbon neutrality. The goal of this article's research is to study the carbon-neutral sustainable digital marketing model. The traditional price prediction algorithm for carbon trading has too many defects. Therefore, from the perspective of carbon neutrality, this paper aims to use the BPNN algorithm to design a price prediction for carbon trading to assist in the study of sustainable digital marketing models.

In recent years, digital marketing has become an important advertising method, playing an important role in product promotion and brand awareness. There is a significant amount of research on digital marketing. Based on advanced market conceptualization and the emergence of market-shaping strategies and hybrid business models, Ts, A. formulated the theoretical definition and practical solutions of digital marketing for the hybrid business models and market-shaping strategies of national enterprises [1]. Mcfarquhar, G. conducted a digital marketing analysis on the products of optical companies. Using a comparative analysis with traditional sales, he came to the conclusion that digital marketing solutions can increase the sales of optical companies' products by more than 10% [2]. Taboubi, S. studied the impact of digital marketing on product pricing. Technological developments have facilitated the digitization of products and the introduction of various mobile devices designed to consume this digital content [3]. The purpose of a study by Angel, M, M. was to highlight the central role of banking customer engagement as a mediating variable between customer experience and two non-transactional customer behaviors: advocacy and attitudinal loyalty [4]. Research by Gabrielli, J. explored the measurement of alcohol marketing exposure across channels and whether cumulative recall exposure is independently associated with underage drinking [5]. However, their research only compared the difference between digital marketing and traditional marketing for product sales, and did not conduct in-depth research on digital marketing models.

Carbon neutrality is a common international goal and trade. Many scholars have conducted research focusing on the goal of realizing carbon neutrality. Tattini, J. contributed to the scientific literature by analyzing the long-term decarbonization of the Danish transport sector using energy, economic, environmental and engineering models [6]. Chemical neutralization of carbon dioxide was analyzed by Fukushima, T., who demonstrated a carbon-neutral energy cycle using glycolic acid or oxalic acid redox pair [7]. Noterdaeme, P. has studied a very effective method of finding high metal molecular absorbents with CI-containing absorbents, and the absorbents he studied can effectively alleviate the expansion effect of carbon dioxide [8]. Tew, D. E. described the overall integrated synergy that can be achieved in a hybrid system consisting of a solid oxide fuel cell and an engine bottom cycle [9]. Zhang, S. C. analyzed the contribution of near-zero energy building standards to China's 2060 carbon emission target in urban areas by combining a carbon emission model with a bottom-up mid and long-term energy consumption model [10]. However, their research only focused on carbon dioxide emissions and did not pay attention to the impact of energy replacement on carbon neutrality.

The main innovations in this paper are as follows: In recent years, this paper selected a new type of ground energy as the preferred energy source for carbon neutrality goals, which is less concerning to scholars. Ground energy is secondary energy and clean energy for building air conditioning systems, which is of great significance for energy conservation and emission reduction.

## 2. Geoenergy Resources Promote Carbon Neutrality

### 2.1. Carbon Neutrality Goals

The carbon in "carbon peak and carbon neutrality" refers to the carbon dioxide emitted in the process of social production, which is a long-term process. As a new form of environmental protection, it has been adopted by more and more large-scale activities and conferences that promote green lifestyles and production and realize the green development of the whole of society. In social production, carbon emissions enter a period of rapid or moderate growth early on. China plans to reach the peak of its total domestic carbon emissions by 2030, entering a plateau period for peak carbon emissions. During the platform period, the social carbon emissions will fluctuate within a certain range, but the total amount will remain unchanged and the platform period will be longer or shorter. China plans to gradually achieve a rapid or slow decline in overall social carbon emissions after the plateau period by means of industrial structure upgrading, supply-side reform, regional economic integration, ecological nature protection, and energy utilization efficiency improvement. Ultimately, before 2060, efforts should be made to achieve the neutralization of carbon emissions in the whole industry; that is, to offset carbon emissions in social production and achieve the goal of "net zero emissions" [11], as shown in Figure 1.

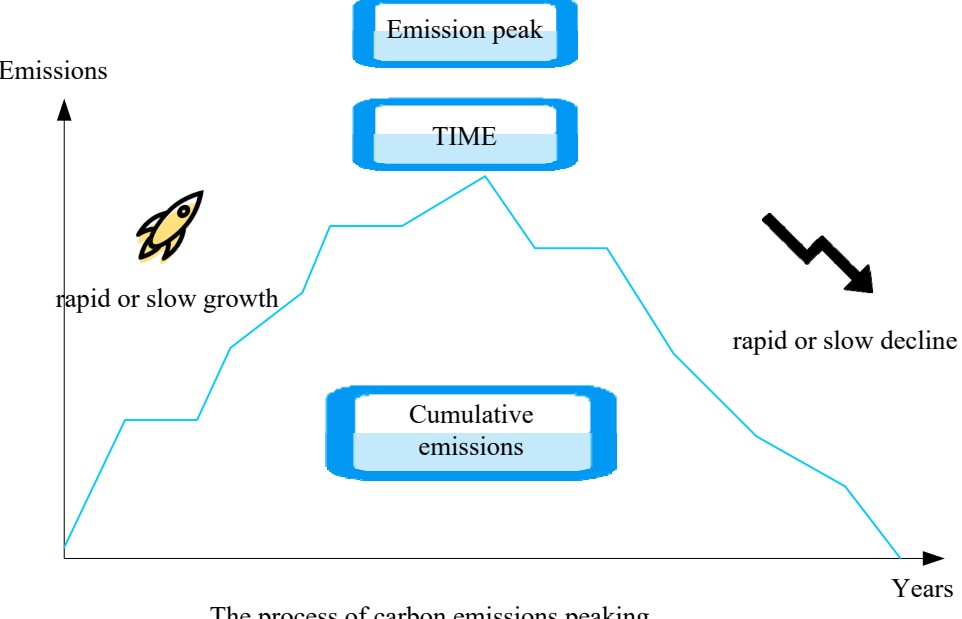

**Figure 1.** Carbon Peaking Process.

"Carbon peaking and carbon neutrality" is a complex situation due to which China currently faces international economic uncertainty and instability. As the main secondary energy in today's society, the power industry is closely related to "carbon peaking and carbon neutrality". The State Grid Corporation has also actively responded to the call of the state; in combination with the "14th Five-Year Plan", they have conducted in-depth research and specific planning and deployment to meet the "carbon peaking and carbon neutrality" goal [12].

With the introduction of the concept of comprehensive energy, constructing energy Internet, solving the environmental problems caused by the gradual depletion of fossil fuels, realizing the open interconnection of multiple energy sources, and realizing the cascade utilization of energy have become research hot spots. Power grid companies have the ability, responsibility and obligation to assist the government in researching and promoting the optimal scheduling of user-side distributed energy under the framework of the energy Internet. It needs to promote scientific planning for energy development,

enhance the coordinated operation of various energy systems, and help user-side energy achieve optimal energy efficiency, as shown in Figure 2.

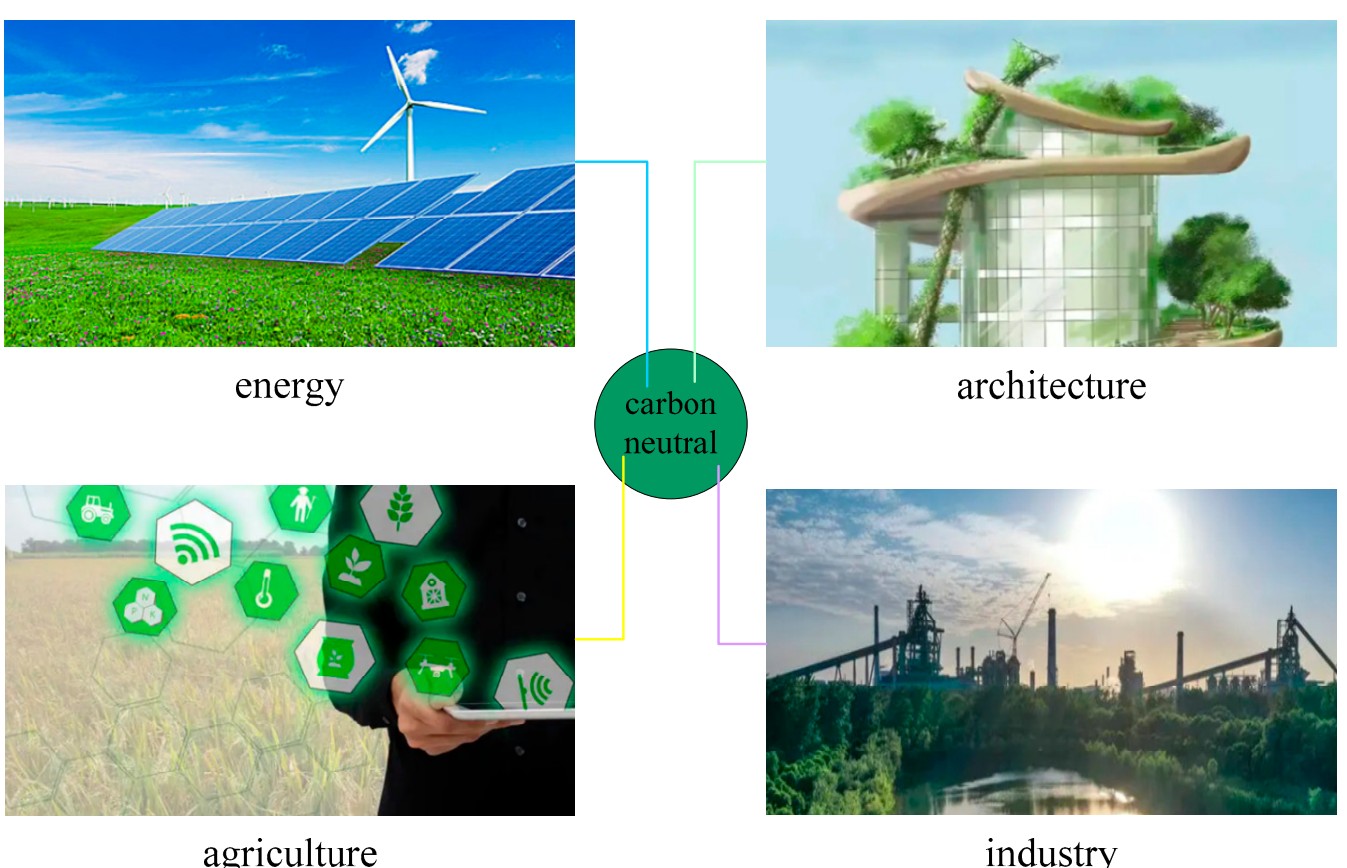

**Figure 2.** Carbon Neutrality.

Community-level user-side distributed energy system, as the basic product of intelligent building platform technology promotion, is very important. It uses the latest technologies such as the integrated Internet, big data, and the Internet of Things, and takes building energy management as the core to scientifically select and standardize energy consumption control and management schemes to realize building intelligence, and achieve the comprehensive effect of energy conservation and emission reduction. It improves building quality while maintaining building comfort, flexibility, safety and reliability. Currently, various low-carbon solutions are employed for building energy supply through various distributed energy systems distributed in buildings [13]. Its basic structure is shown in Figure 3.

Through the coordinated management and optimization of various energy sources, the total energy consumption of the building can be reduced, the comfort of users and the full utilization of renewable energy can be ensured, and the operating costs of the microgrid can be reduced to a certain extent [14].

### 2.2. Utilization of Geoenergy Resources

Today, when energy conservation and emission reduction are advocated, ground energy is growing rapidly in China at a rate of about 10% every year as a new form of clean and renewable energy, and has been widely promoted and applied around the world [15]. It has bright development and application prospects.

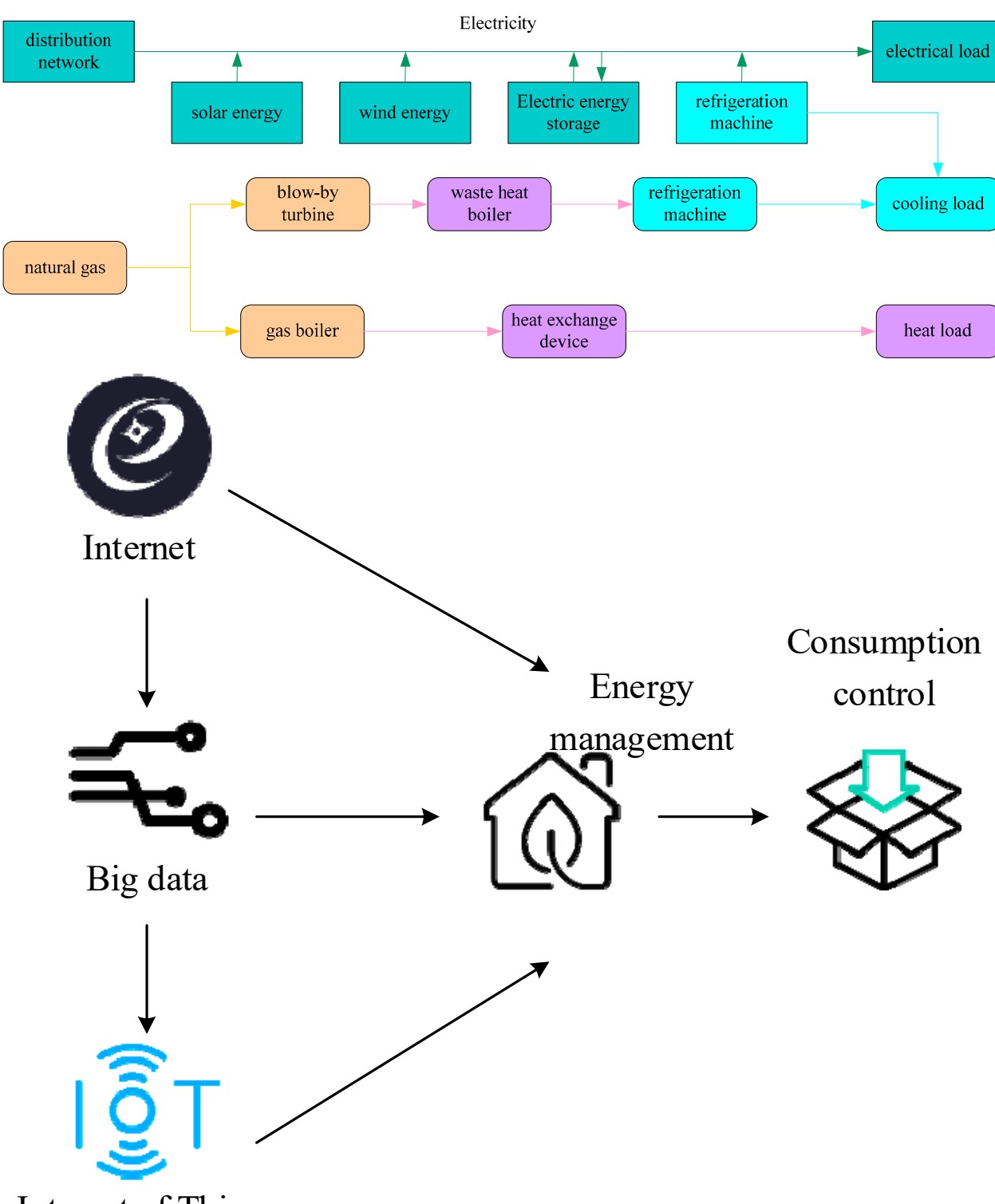

**Figure 3.** Basic Structure Diagram.

The ground-source heat pump system is a new type of environmental protection energy utilization system that uses shallow ground heating and cooling technology. The ground energy heat pump system is the backbone of the sustainable digital marketing model, and its position in sustainable development is very important. As one of the large fixed reserves and stable new energy forms in the world, ground energy has great development potential and has been vigorously promoted in fields such as power generation,

building heating and cooling. Generally, the utilization of geoenergy is divided into three utilization types: shallow geothermal energy, mid-deep hydrothermal geothermal energy and hot dry rock [16]. Shallow geothermal energy has many advantages, such as wide area, large reserves, strong regeneration capacity, and high utilization efficiency. It can be extracted and utilized through heat pump technology, which can not only satisfy the heating and cooling of buildings, but also reduce carbon dioxide emissions and weaken the impact of greenhouse effects [17], as shown in Figure 4.

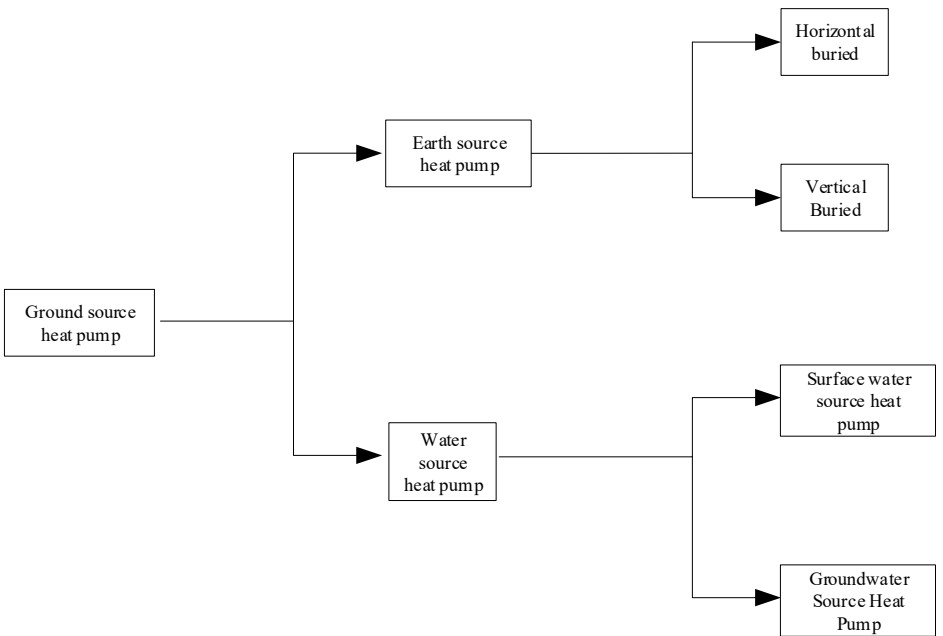

**Figure 4.** Basic Categories of Geoenergy Resources Heat Pump Systems.

As shown in Figure 4, although geoenergy resources heat pump technology has achieved good economic and social benefits, with the development of its large-scale application, a series of problems involving the depletion of ground energy resources and the lack of research on complex underground thermal rheological processes have become increasingly prominent, which limits the further promotion of ground energy utilization technology. At the same time, the possible negative effects of the large-scale application of ground source heat pumps on the ecological balance of underground heat and the environment also require in-depth investigation and research. In order to realize the stable and sustainable operation of the system and realize the dynamic balance of the underground thermal environment, the key is to conduct a comprehensive and systematic investigation and research on the underground heat exchange part. It is necessary to deeply analyze the controllable factors affecting the temperature field, explore the law of underground heat transfer, realize the optimal design of the system, and maintain the dynamic balance of the geothermal field [18]. Due to today's increasingly scarce resources, heat pump technology, as the most advanced way of utilizing low temperature heat energy, has been widely adopted and valued by countries around the world.

In the development process of ground energy utilization heat pump technology, the problem of heat penetration is an important factor affecting its operation stability, efficiency and life. Excessive heat penetration would reduce system efficiency and even accelerate system failure. Therefore, research aiming to solve the heat penetration problem is the key to improve the utilization efficiency of ground energy. In order to realize the accurate prediction of the change trend of thermal penetration, it is necessary to grasp the change law of the ground temperature field. The reality is that research conducted by China and other countries has not focused on this aspect, instead giving their attention to the ground. Because underground heat exchange is not visible, it is difficult to arrange points

for observation. At the same time, the heat transfer of underground porous soil involves complex heat flow processes such as conduction, convection, and dispersion, which also causes certain difficulties in the study of underground heat transfer. However, in order to realize the sustainable and healthy operation of the ground source heat pump, the underground heat transfer mechanism must be deeply explored, and the dynamic control of the evolution process of the geothermal field must be realized to effectively serve the engineering practice [19].

As a good form of renewable energy, shallow underground energy has significant energy storage characteristics, so the efficient utilization of ground energy resources has always been of high concern. It is of great significance to improve the energy transmission performance of ground-source structures, realize the effective recycling of energy resources and deeply study various forms of underground energy transmission processes. For instance, shallow geothermal energy refers to the low-temperature heat energy contained in the soil sand and groundwater within hundreds of meters of the shallow surface of the earth.

In ground source and ground energy utilization technology, the ground source heat pump is an efficient and environmentally friendly air conditioning system that typically utilizes shallow geothermal resources for building cooling and heating, as shown in Figure 5.

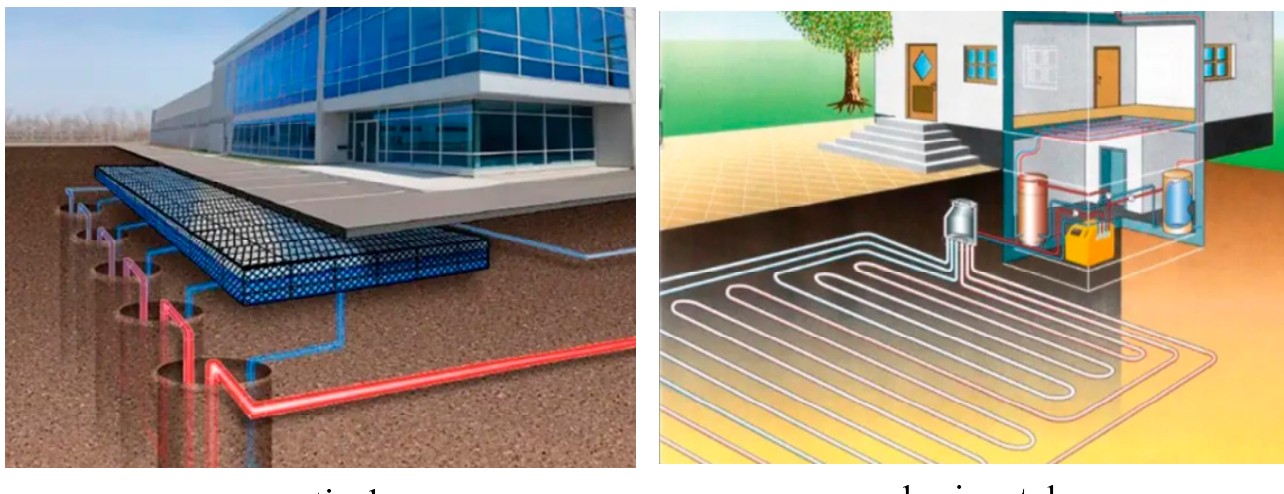

vertical          horizontal

**Figure 5.** Principles of Two Geoenergy Resources Heat Pumps.

In the process of exploring geothermal exploitation and utilization, various localities have also found new methods of urban energy transformation. In the typical application of shallow ground energy utilization, the form of underground heat exchange is usually divided into a buried tube underground heat exchanger and pumping well, as shown in Figure 6. The buried tube heat exchanger system is a closed loop pipeline that utilizes the circulating fluid in the pipeline, indirectly realizes the heat exchange between the fluid and the soil through the heat conduction between the pipe wall and the soil, and realizes the extraction and utilization of geothermal energy through the heat pump heat exchanger. A different well pumping-irrigation underground heat exchange system is equipped with a pumping well and recharge well. The groundwater is directly extracted through the water filter pipe of the pumping well and enters the heat exchanger of the heat pump unit. After heat exchange with the heating (cooling) space is achieved, it is injected into the aquifer again from the recharge well to maintain the stability of the groundwater dynamic field. Because the system directly utilizes groundwater with higher specific heat capacity as the heat exchange medium, it has higher operating efficiency than the buried tube heat exchanger system [20].

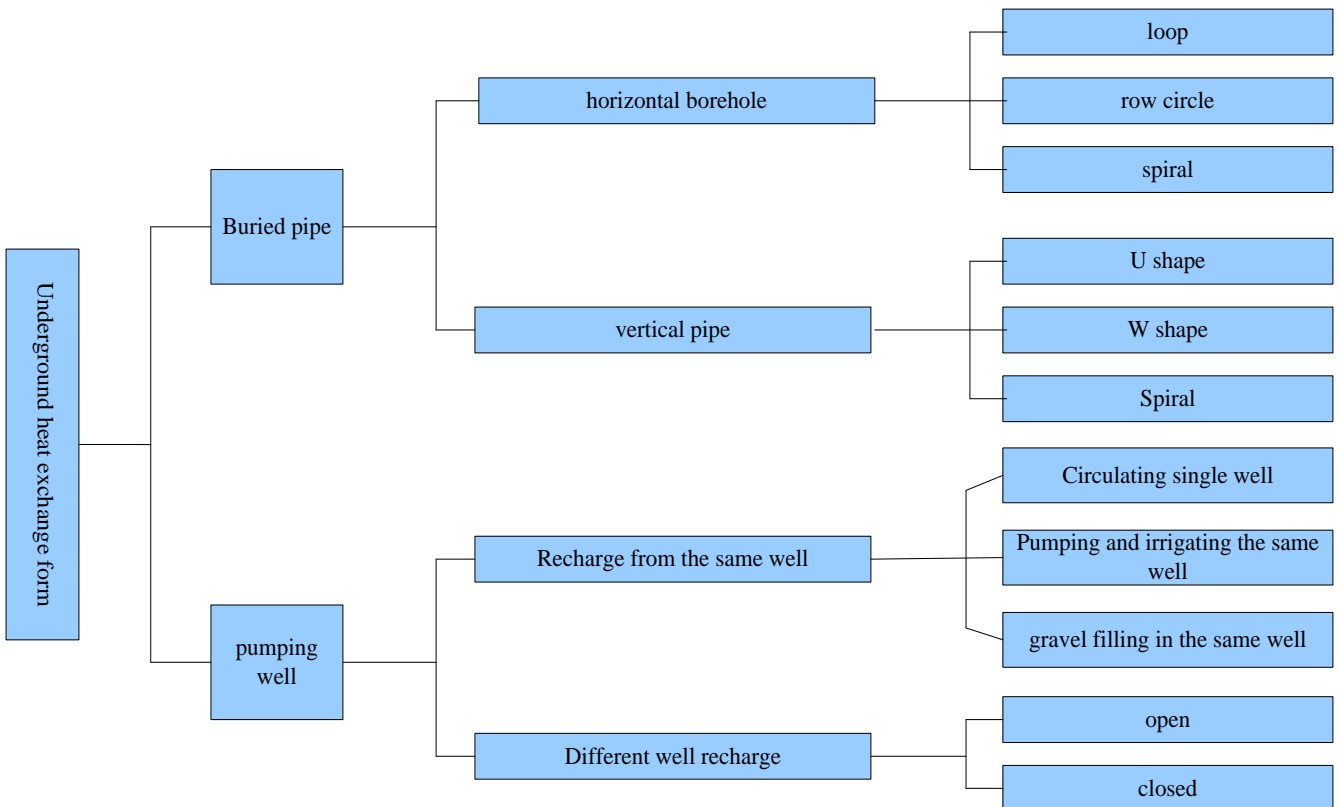

**Figure 6.** Basic Types of Underground Heat Transfer Forms.

*2.3. Carbon Trading Price Prediction Algorithm Based on BPNN*

This article uses the BPNN algorithm to build a sustainable digital marketing model to help it predict the price of the carbon trading market to promote the push of the digital marketing model.

Carbon trading price prediction model of BPNN model: First, the optimal mixed models based on energy, economy, weather and environmental factors are established, respectively. Then, the optimal mixed models are combined to predict the carbon trading price and obtain the initial prediction error. Then the BPNN model is used to predict the initial error to obtain the error prediction value. Finally, the carbon trading price prediction value after error correction is obtained. The prediction results show that the model can effectively improve the prediction accuracy of the carbon trading price model.

The input and output data need to be preprocessed and reverse preprocessed. This is because there is a big difference between the values when the impact factor index is used as the input and the carbon emission right price is used as the output. After preprocessing, the input data and output data can be controlled within a certain range. In order to provide good training conditions for the BPNN, the value is usually converted into the 0–1 interval; that is, the samples are normalized. The preprocessing formulas are:

$$X_{ij} = \frac{x_{ij} - x_j^{\min}}{x_j^{\max} - x_j^{\min}} \tag{1}$$

$$x_j^{\max} = \max_{i=1}^{n}\{x_{ij}\} \tag{2}$$

$$x_j^{\min} = \min_{i=1}^{n}\{x_{ij}\} \tag{3}$$

In these formulas, $X_{ij}$ is the value input to the neural network after preprocessing, $x_{ij}$ is the original value of the collected influencing factors. $x_j^{\min}$, $x_j^{\max}$ are the minimum and maximum values of these types of influencing factors, respectively.

After the input and output are determined, the neural network can be trained. The specific BPNN training process is as follows:

Through the above analysis, it can be determined that: In the model established in this paper, the number of nodes in the input layer is 6, the number of nodes in the output layer is 1, and the number of hidden layers is 1.

Usually, the number of hidden layer nodes *L* has the following three reference formulas:

$$L = \sqrt{M + N} + a \tag{4}$$

$$L = \log_2 M \tag{5}$$

$$L = \sqrt{MN} \tag{6}$$

Among them, *a* is a constant between 1 and 10. The number of hidden layer nodes finally determined in this paper is $L = 15$.

With the help of the formula, the hidden layer output *H* is calculated by using the input vector *X*, the connection weight $w_{ij}$ between the input layer and the hidden layer, and the hidden layer threshold a.

$$H_j = f\left(\sum_{i=1}^{n} w_{ij} x_i - a_j\right), j = 1, 2, \cdots, L \tag{7}$$

*f* is the activation function, and the function must meet the requirements of continuous differentiability in the setting. Generally, functions such as *logsig*, *tansig*, and *purelin* are selected:

$$\log sig(x) = \frac{1}{1 + e^{-x}} \tag{8}$$

$$\tan sig(x) = \frac{2}{1 + e^{-2x}} \tag{9}$$

$$purelin(x) = x \tag{10}$$

The predicted output *O* of the BPNN needs to be calculated by the following formula, and the parameters used include: the hidden layer output *H*, the connection weight $w_{jk}$ and the threshold value b.

$$O_k = \sum_{i=1}^{L} H_i w_{ik} - b_k, k = 1, \cdots, m \tag{11}$$

When using BPNN to make a causal judgment between independent variables and dependent variables, the observed values of independent variables and dependent variables are used as the input and output of BPNN, respectively. When there is a discrepancy between the output of the neural network and the expected output, the difference is obtained to obtain the size of the error. When the error is within the unacceptable range, the BPNN model would use its own error back-propagation algorithm when training the neural network, and the BPNN would not stop training until the error enters an acceptable range.

## 3. Carbon Trading Market Price Forecast for Geoenergy Resources

### 3.1. Data Sources and Data Processing

By consulting relevant literature, in recent years, the changes in carbon emission trading prices in pilot provinces and cities have been grasped, and the quantification difficulty of each variable index and the difficulty of data collection have been comprehensively evaluated. When studying the variables that affect the trading price of carbon emission rights, this paper mainly selected seven influencing factors, namely coal price, crude oil

price, natural gas price, air quality index (AQI), EUA futures settlement price, CSI 300 index, and SSE Industrial Index, as shown in Table 1. The specific data sources are described as follows.

**Table 1.** Explanation of the Meaning of the Indicators.

| Serial | Index | Indicator Meaning | Indicator Source |
|:---:|:---:|:---:|:---:|
| 1 | X1 | The price of geoenergy resources | Wind Database |
| 2 | X2 | traditional energy prices | Wind Database |
| 3 | X3 | new energy prices | Wind Database |
| 4 | X4 | Air Quality Index (AQI) | 2345 Weather Forecast |
| 5 | X5 | EUA futures settlement price | Wind Database |
| 6 | X6 | CSI 300 Index | Wind Database |
| 7 | X7 | SSE Industrial Index | Wind Database |
| 8 | Y | carbon emissions trading price | China Emissions Trading Network |

### 3.1.1. Price of Carbon Emission Rights

Carbon emission rights are valuable assets that can be exchanged as commodities in the market. The research object of this paper is the carbon emission trading price. Therefore, it is used as the dependent variable; that is, the output value of the BPNN. The selected sample object is the carbon emission rights trading price, and the selection time is the valid working day data from 10 July 2019 to 10 July 2021.

### 3.1.2. Energy Price Indicators

For power enterprises, in the past two years, in the high coal price and the stagnation of coal and electricity linkage under the effect, the profits of the power industry are generally not optimistic. According to information released by relevant departments in 2018, traditional energy consumption accounted for the highest proportion of China's energy consumption, accounting for nearly 80%. New energy consumption accounted for only 14%, and ground energy resources accounted for less than 10%. The accounting subjects of the industrial industry are electricity, steel, cement, glass, etc., and the main sources of greenhouse gases are coal and diesel. Based on the above situation, this paper considered the price factors of traditional energy when measuring the energy price index. Secondly, as a clean energy, natural gas is more environmentally friendly than coal, so natural gas is used more and more frequently. Thus, the price of natural gas is also taken into account in this article.

### 3.1.3. Climate and Environmental Indicators

The climate index is the characteristic quantity of a single climate element or a variety of climate elements under certain climate conditions. With the emission of greenhouse gases, air quality has attracted the attention of many experts and scholars. In order to avoid subsequent environmental deterioration, cities with good air quality are likely to increase the price of carbon emission rights to achieve the purpose of steadily improving air quality and improving urban living standards.

### 3.1.4. Macroeconomic Indicators

As the economic situation improves, the production scale of enterprises continues to expand, and carbon emissions are also increasing. The CSI 300 Index was jointly issued by

the Shanghai Stock Exchange and the Shenzhen Stock Exchange on 8 April 2005, including the prices of stocks in Shanghai and Shenzhen. The stocks selected into the index are all highly liquid and large-scale stocks, which can fully reflect the operation of China's overall economy. Therefore, the CSI 300 index in this article is selected to represent the macroeconomic indicators. The data come from the Wind database and the time period is the same as above.

### 3.1.5. Industrial Development Indicators

Industrial and chemical-related companies are the largest demanders of carbon emission rights, and relevant studies have shown that the transaction price of carbon emission rights is linked to industrial conditions. If the number of industrial enterprises increases, more greenhouse gases would be produced, and the demand for carbon emission allowances for the whole society would also increase proportionally. According to the price determination theory, when the demand is greater than the supply, the price of carbon emission rights would rise. In addition, industrial production is also closely related to energy demand. After the scale of industrial production is expanded, the energy required to consume increases, and eventually a large amount of carbon dioxide is emitted. All in all, when the number of industrial companies included in the scope changes, it changes the overall demand for carbon emission rights, which in turn has an impact on prices. Therefore, this paper chose the Shanghai Stock Exchange Industrial Index as an industrial indicator. The data come from the Wind database and the time period is the same as above.

A total of 500 sample data were collected and summarized. Due to the limited length of the article, only some relevant data are selected below and some of the results are shown in Table 2.

**Table 2.** Some Indicator Data.

| TIME | X1 | X2 | X3 | X4 | X5 | X6 | X7 |
|------|------|------|------|----|------|--------|--------|
| 7.8 | 5.88 | 6.29 | 4.96 | 60 | 2.86 | 3797.2 | 2081.3 |
| 7.9 | 5.90 | 6.29 | 4.95 | 61 | 2.65 | 3793.6 | 2083.7 |
| 7.10 | 5.83 | 6.29 | 4.95 | 58 | 2.68 | 3826.3 | 2090.3 |
| 7.11 | 5.82 | 6.40 | 4.96 | 35 | 2.63 | 3896.4 | 2145.6 |
| 7.12 | 5.85 | 6.42 | 5.02 | 41 | 2.57 | 3893.7 | 2147.3 |

### 3.2. Carbon Emissions Trading Price Prediction Based on BPNN Model

#### 3.2.1. Multiple Regression Prediction

In the process of literature research, it was found that most scholars in the past chose econometric models for research on such issues. Therefore, before the BPNN test, this paper first conducted multiple regression analysis on the sample data. Firstly, it can study the influence degree of the influencing factors selected in this paper on the price of carbon emission rights, and, secondly, it can compare the empirical results with the BPNN model in the following. This paper used Eviews10 for regression analysis. A linear regression model was established and the results are shown in Table 3.

From the data in Table 3, it can be seen that the coefficients before X1, X4 and X6 are negative, indicating that the coal price, AQI and CSI 300 index of Zhengzhou Commodity Exchange are negatively correlated with the price of carbon emission rights. The remaining coefficients before X2, X3, X5, and X7 are positive, indicating that the retail price of diesel in Wuhan, the retail price of natural gas in Wuhan, the settlement price of EUA futures and the Shanghai Stock Exchange Industrial Index are positively correlated with the price of carbon emission rights. In addition, the *p* values of X1, X2, and X4 are all greater than 0.05, and the regression is not significant, indicating that the coal price of Zhengzhou Commodity Exchange, the retail price of diesel in Wuhan, and AQI have no significant linear effects on carbon prices.

**Table 3.** Linear Regression Results.

| Variable | Coefficient | Std. Error | t-Statistic | Prob |
|----------|-------------|------------|-------------|------|
| C | −4.820 | 7.369 | −0.654 | 0.513 |
| XI | −0.00 | 0.009 | −0.279 | 0.779 |
| X2 | 1.185 | 0.758 | 1.563 | 0.118 |
| X3 | 0.584 | 0.207 | 2.813 | 0.005 |
| X4 | −0.009 | 0.005 | −1.857 | 0.063 |
| X5 | 1.179 | 0.048 | 24.293 | 0.000 |
| X6 | −0.008 | 0.001 | −4.468 | 0.000 |
| X7 | 0.012 | 0.003 | 3.625 | 0.000 |

The determination coefficient of the model $R^2 = 0.822623$ shows that the model fits the sample data well on the whole; that is, the explanatory variables explain the explanatory variables to a large extent. However, when the significance level $\alpha = 0.05$, t1, t2 and t4 are less than the critical value, it indicates that the model may be disturbed by multicollinearity. Therefore, in-depth conclusions can be drawn from the correlation coefficient matrix. The results are shown in Table 4.

**Table 4.** Correlation Coefficient Matrix.

| | X1 | X2 | X3 | X4 | X5 | X6 | X7 |
|----|----|----|----|----|----|----|----|
| X1 | 1 | 0.22 | 0.3232 | 0.0234 | −0.4462 | 0.3905 | 0.3316 |
| X2 | 0.22 | 1 | 0.1721 | −0.0472 | 0.482 | −0.3506 | −0.631 |
| X3 | 0.3232 | 0.1721 | 1 | 0.4431 | −0.0041 | 0.1502 | 0.0442 |
| X4 | 0.0234 | −0.0472 | 0.4431 | 1 | −0.0881 | 0.1176 | 0.1349 |
| X5 | −0.4462 | 0.482 | −0.0041 | −0.0881 | 1 | −0.5265 | −0.7422 |
| X6 | 0.3905 | −0.3506 | 0.1502 | 0.1176 | −0.5265 | 1 | 0.9024 |
| X7 | 0.3316 | −0.631 | 0.0442 | 0.1349 | −0.7422 | 0.9024 | 1 |

It is not difficult to see from the correlation coefficient matrix that the correlation coefficients between some explanatory variables are very high, indicating that the model has a multicollinearity problem. Therefore, stepwise regression is used to obtain the optimal regression model:

$$\hat{y} = 0.4326X3 + 1.154X5 - 0.005X6 + 0.008X7 + 5.159 \tag{12}$$

A prediction of the price of carbon emission rights is carried out through this new model and the results are shown in Figure 7.

As shown in Figure 7a, the overall fitting degree is not high, and there are obvious deviations in the data 13, 14, and 15. The specific error value is shown in Figure 7b, the error value is between −6 and 7 and the maximum error reaches 34%.

### 3.2.2. BPNN Prediction

Next, the BPNN model selected in this article is used. In this paper, the parameters of the model have been determined and the data have been processed. Then, the MATLAB toolbox can be used to run the program, train and simulate the data, and predict the value of carbon emission rights.

First, 450 samples (90% of the total) that are the same as the previous section were selected as the input of the neural network. After that, the initial simulation and training learning of the BPNN were carried out in the MATLAB toolbox until the average error was

less than the value set when the model was established at the beginning. At this point, the BPNN would stop training. Then, 30 sample data (accounting for 6% of the total) were selected for verification. At the end, the system would generate line graphs such as prediction error percentage, expectation and prediction output comparison. The training results of the model are shown in Figure 8.

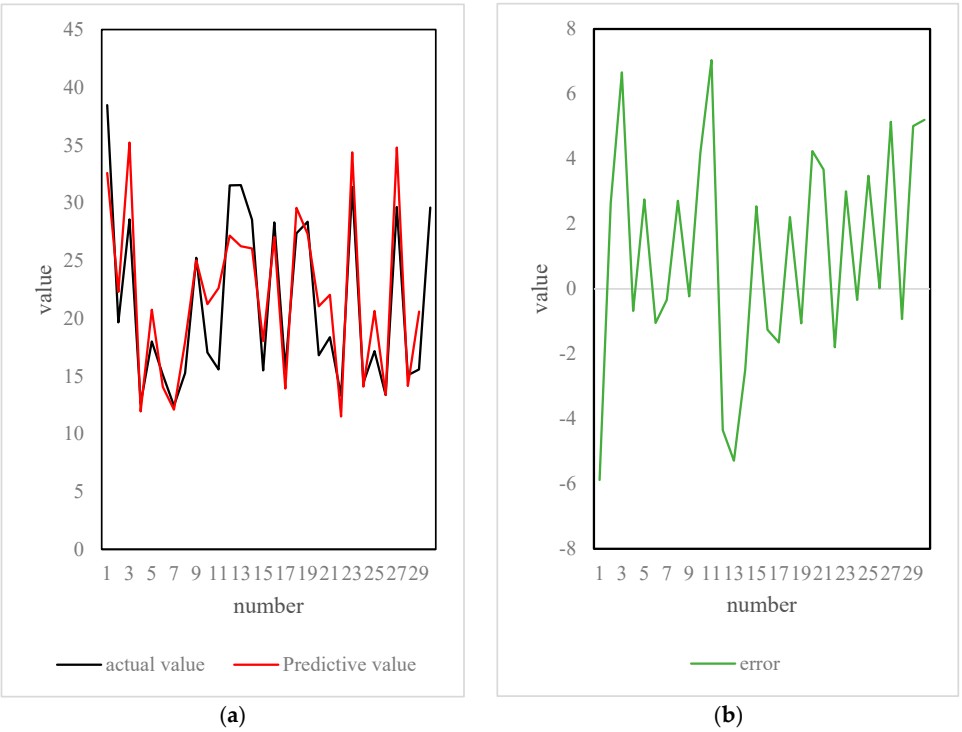

(**a**)                      (**b**)

**Figure 7.** Multiple Regression Prediction Results: (**a**) Comparison of Actual Value and Predicted Value; (**b**) Error value.

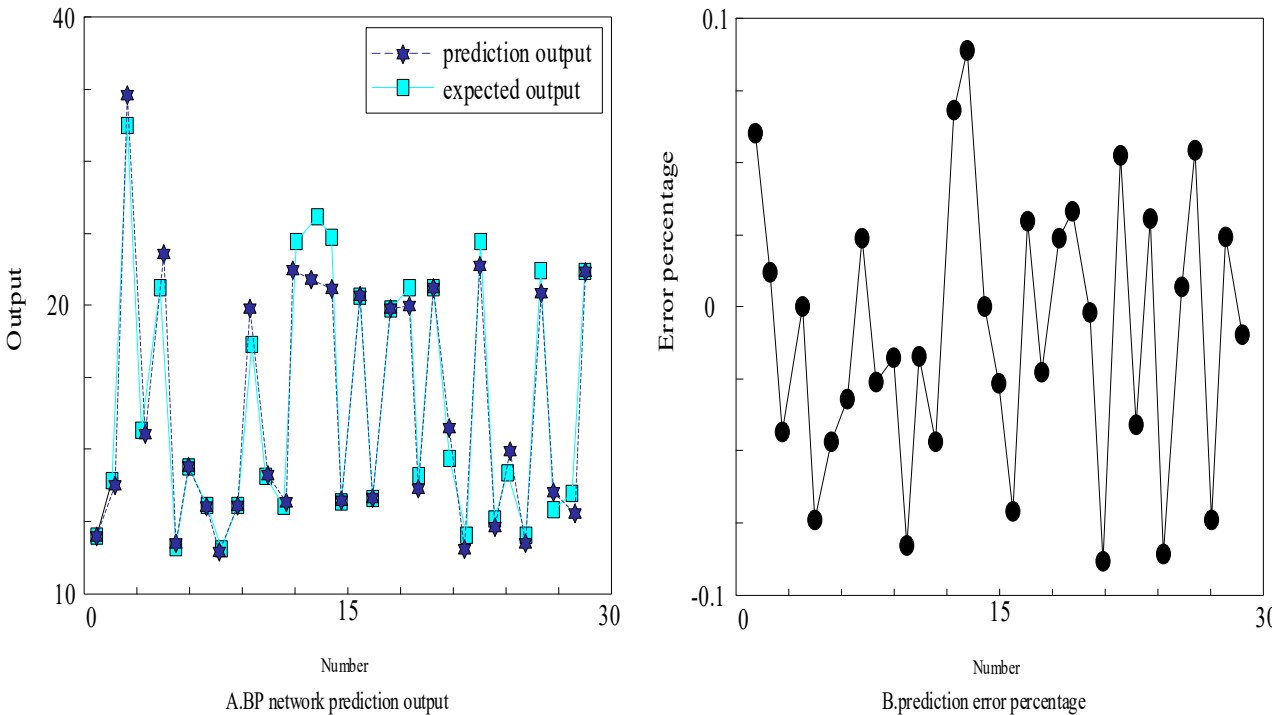

**Figure 8.** Model Training Results.

As shown in Figure 8A, the overall fitting degree is high. The predicted value is not much different from the output value, and only the three samples of 12, 13, and 14 have a slight deviation. The specific error rate is shown in Figure 8B. The overall error is within 10%, and the average error is about 5%, indicating that the model works well.

The quality of the judgment model results is usually measured by two indicators: mean square error (MSE) and goodness of fit ($R^2$). The smaller the value of MSE, the closer the value of $R^2$ is to 1, indicating that the closer the predicted value is to the expected value, the better the fit of the model is. The model goodness of fit was 0.0989, and the model mean square error was 1.370. It can be seen from these two indicators that the current BPNN model has a high degree of fitting. However, in order to further study the stability of the model, this paper introduced a cross-validation model, which can further evaluate the pros and cons of the established BPNN. Its basic principle is to divide the sample data into several groups on average. Each group is used as a validation set in turn, and the samples of each remaining group are used as a training set, and then the average value of each validation error is calculated as an evaluation index.

### 3.2.3. Comparison of the Effects of the Two Models

The following article compares the error value of the BPNN model with the error value predicted by the multiple regression model. The comparison results are shown in Figure 9:

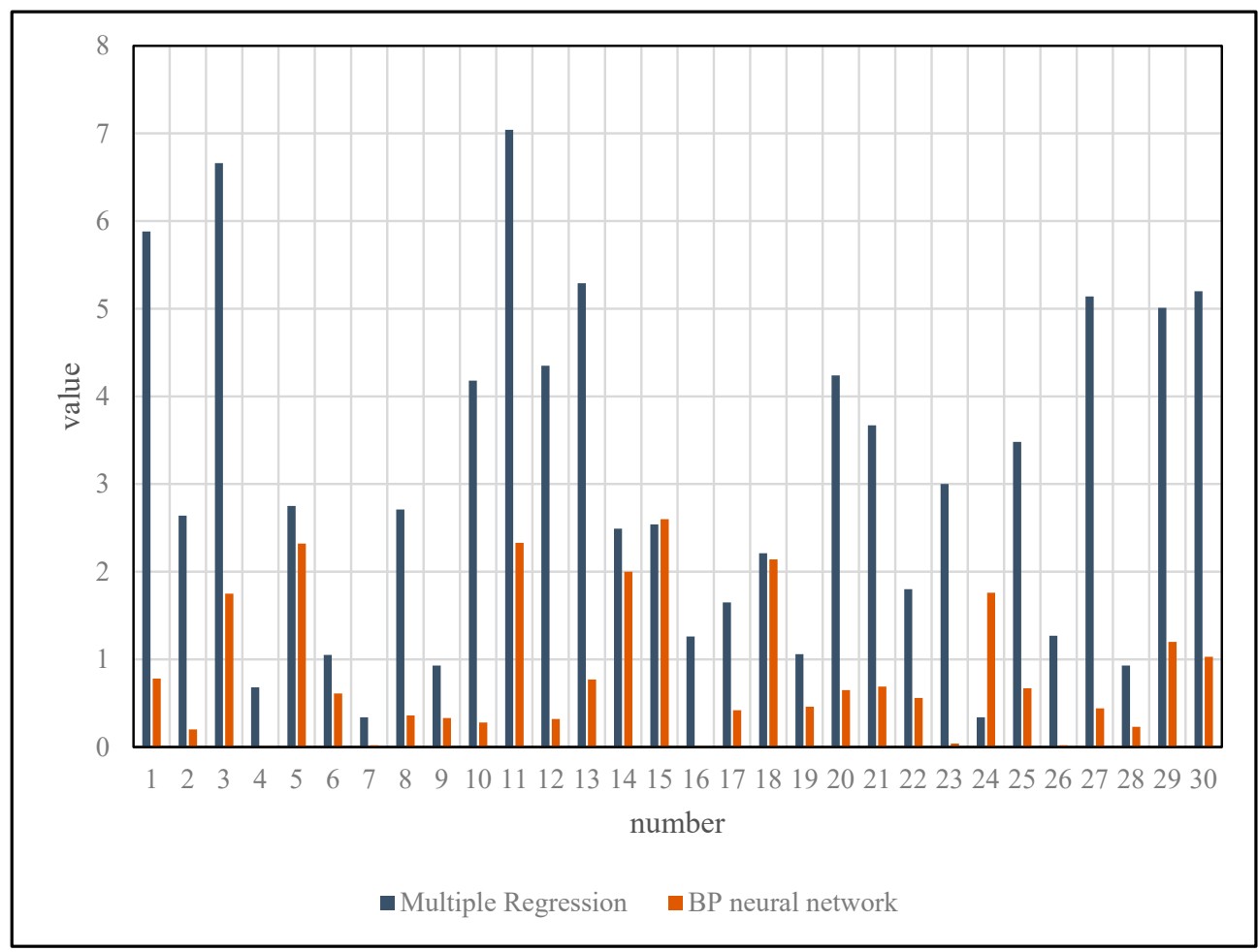

**Figure 9.** Prediction Difference.

As can be seen from Figure 9, the BPNN has a smaller predicted error value than the multiple regression. At the same time, in order to make the results more intuitive, the mean

square error (MSE) value predicted by the multiple regression model was also calculated to be 12.5022. The MSE of the BPNN was only 1.3696.

Accordingly, through the above test and analysis, it can be concluded that the BPNN has better generalization and simulation capabilities. It is more accurate than the multiple regression prediction, and can be used to predict the price of carbon emission rights in Hubei Province. Based on this as a reference, the price of carbon emission rights in other provinces in the country can also be estimated and analyzed.

Applying BPNN analysis to the carbon emission trading price forecast has both its own advantages and some shortcomings. First, its advantages are: For Hubei Province, the trading volume of carbon emission rights is large, the price samples are very sufficient and easy to collect, and the data are authentic and credible. As long as the structure of the BPNN is properly set, it can play a black-box role and analyze the relationship between the variables. By checking the degree of fit between the input and output of the model, it is possible to analyze whether the test effect, simulation and generalization performance of the network are good, and high accuracy can also be obtained when predicting the effect of similar samples in the future. Secondly, for the study of carbon emission rights trading price and its influencing factors, the BPNN is different from the traditional mathematical model to find the correlation between variables from a linear perspective. In fact, it is rare that the relationship between variables in life is simple and linear. If the non-linear relationship is reduced to a linear relationship with the help of statistical analysis methods, it definitely leads to errors. The advantage of BPNN is that it learns and simulates variables on the basis of a complete database. The output of the neural network after successful training does not simplify the nonlinear relationship, so it can better reflect the actual situation. Finally, in this paper, BPNN is used for parallel computing, which is very suitable for many complex factors that affect the price of carbon emission rights. At this level, BPNN is naturally conducive to estimating the price of carbon emission rights. In addition, the BPNN model can automatically check out the nonlinear relationship between the carbon emission trading price and other potential influencing factors from a large number of training samples, so that the research would not fall into empirical and preconceived misunderstandings.

## 4. Sustainable Digital Marketing for Carbon Markets

### 4.1. Digital Marketing of Carbon Trading

Digital marketing is the practice of using digital communication channels to promote products and services to communicate with consumers in a timely, relevant, customized and cost-effective manner. Digital marketing encompasses many of the techniques and practices found in Internet marketing (network marketing). The scope of digital marketing is broader, including many other communication channels that do not need the Internet, such as television, radio, SMS and other non-Internet channels, or social media, electronic advertising, banner advertising and other online channels.

In the wave of digitization, it can be said that binary 0 or 1 dominates the Internet world. A lot of complex information can be converted into digital form and processed efficiently and uniformly with the help of computer technology. This digital form enables frequently encountered pictures, texts, voices, videos, etc., to be quickly disseminated and exchanged on the Internet through standardized digital codes, breaking through the limitations of space.

Before mentioning the development of digital marketing, the development and evolution of technology must be explained in detail. The update and iteration of technology is the infrastructure for the development of digital marketing. The three development stages of PC Internet, mobile Internet and IOT are relatively well-known. Web 1.0 and web2.0 are actually terminal computers in the PC Internet stage, and web3.0 stage is the transition of the integration of PC, mobile Internet and Internet of Things. The network forms of three different forms and different carriers go hand in hand. The era of the Internet of Everything has come quietly, and the comprehensive development of big data, cloud computing, AR and other technologies has laid a solid technical foundation for the new era of intelligence.

The development of digital marketing is also relatively clear. At the beginning, it was mainly traditional online advertising, then social network marketing, and then the more prosperous mobile marketing and intelligent marketing stages. The marketing methods at each stage have different characteristics and emphasis. With the development of digital technology, digital marketing has gradually become the most important marketing method. On the Internet, digital marketing seen by ordinary people exists in the form of rich media advertising, information dissemination and collection. However, it is more reflected in the result of complex and precise operations of virtual numbers, which represent the precise crowd placement operation strategy.

### 4.2. Need for Digital Marketing

With the sweeping wave of the industrial Internet, the application of big data technology is becoming more and more mature, which has also introduced people to new business ideas, especially the recent COVID-19 pandemic in 2020. The offline business of many traditional enterprises was forced to be completely interrupted, and the flow of economic value was also stagnant, resulting in the closure of enterprises and the unemployment of a large number of employees. The country also suffered huge losses. More importantly, the competition in offline scenarios is becoming more and more intense. The traditional marketing model is not only expensive, but also has a declining profit contribution to the company. In the digital age, the development of various industries is gradually accelerating and the needs of buyers are changing rapidly. This also leaves a lot of questions for enterprises to think about. They are actually faced with marketing confusion and challenges. At this time, data have become an important reference dimension for enterprise decision making, and digital marketing has also become a breakthrough in enterprise marketing and an effective way to obtain marketing dividends. In the era of mobile Internet, human social activities have generated a huge amount of information, which is quickly transmitted through mobile terminals and also brings a large amount of data references to corporate marketing. Moreover, due to the built-in GPS positioning system, gravity sensor, high-definition camera, and wireless transmission technology in mobile phones, mobile smart terminals can realize more comprehensive data mining and sorting. At the same time, it can quickly model and outline the user's Internet portrait in the Internet world. In addition, based on historical browsing records and behavior data, the user's back or potential consumption needs can be estimated. Therefore, the corresponding solutions are provided through the Internet, which also allows enterprises to achieve the purpose of precise marketing according to user needs. The value pursued by advertising has also changed from pure exposure in the past to the dimension of final transaction effect. In the era of digital marketing, it is not simply branding advertising, effective advertising, and pure effect CPC billing model advertising. More and more companies have begun to think deeply from the effect side. Reasonable matching of brand exposure and effective clicks on advertisements can exert the superposition effect of the marketing mix, and the feedback is better at the sales level of the company. At the level of effective advertising, it is dismantled at various levels such as form, channel, pit, and conversion, and then different departments and strategies are derived to continuously optimize and promote the upgrade of digital marketing. The digital economy is becoming an important engine driving the high-quality development of China's economy.

## 5. Discussion

Ground energy assets are another clean energy source. Instead of harvesting energy directly, the earth can harvest heat energy from shallow soil and surface water in a roundabout way. Huge strides for energy assets in China's non-partisan carbon targets have been made in conserving energy and reducing emissions. In any case, the ongoing carbon trading market has not been figured out by the average person. The reason for this paper is to decompose the impact of terrestrial energy assets on advancing manageable computerized advertising models. In order to achieve the goal of reducing carbon emissions, countries

around the world are doing their utmost. This paper explores financial targets for carbon indifference by examining indicators in carbon trading markets. Therefore, this paper intends to start from the perspective of BP brain organization (BPNN) to predict and calculate the costs of carbon trading, so as to predict the costs of the carbon trading market, and then promote the accurate push of the electronic display mode method. Therefore, through the above tests and analyses, it can be concluded that BPNN has good generalization and simulation capabilities. This model is more accurate than multiple regression prediction, and can be used to predict the price of carbon emission rights in Hubei Province. With this as a reference, people can also estimate and analyze the price of carbon emission rights in other provinces across the country.

## 6. Conclusions

The traditional energy price is still the biggest constraint. Due to the inconsistent development of different markets, the market response mechanism is not perfect and the degree of regional economic development is limited. Therefore, the research in this paper finds that in economically developed areas, the price of coal has a positive effect on China's carbon trading, while in economically backward areas, there is a negative effect. Whether traditional energy has a positive or negative effect on changes in carbon trading prices needs to be further explored, but what is certain is that traditional energy plays a significant role in the market. On this basis, the article puts forward several noteworthy issues the following suggestions: First, the scope of the sample is limited. This paper only conducts an empirical analysis of the carbon trading market in Hubei Province within two years. This does not fully reflect the carbon trading market in the whole country. Second, some domestic policy indicators are missing because some data are difficult. Finally, due to the limitation of the option period, there is no guarantee that under certain conditions, there would be no sudden change in the transaction price. Therefore, in future research, an empirical analysis should be conducted on the national carbon trading market. It should be more thoughtful in data handling. Therefore, this paper should be more extensive and careful in terms of data processing in future research.

**Author Contributions:** Y.Z. and Z.X. are responsible for analyzing the data; Y.L., A.D. and J.W. contributed lead writing. All authors have read and agreed to the published version of the manuscript.

**Funding:** This study did not receive any funding in any form.

**Institutional Review Board Statement:** Not applicable.

**Informed Consent Statement:** Not applicable.

**Data Availability Statement:** Data sharing not applicable to this article as no datasets were generated or analyzed during the current study.

**Conflicts of Interest:** The authors declare that there is no conflict of interest with any financial organizations regarding the material reported in this manuscript.

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
