# Peer review of "Sustainable Digital Marketing Model of Geoenergy Resources under Carbon Neutrality Target"

_sustainability, doi:10.3390/su15032015_

Round 1
Reviewer 1 Report
Dear Author(s)
comments
1. Introduction section, novelty is missing.
1 2. Author need to include list of abbreviations; it will make easier for reader to read and understand.
1 3. The conclusion part is also needed to be revised; which questions are answered, what is the value/originality/contribution of the paper, how the proposed method answers the research questions that previous methods are not able to answer.
4. Please propose and suggest more possible future studies related to the current study.
Author Response
- Introduction section, novelty is missing.
Answer: Thank you for your suggestion, I have revised the introduction, I revised it to: This not only reflects the difference in the development of the industry, but also reflects the signs of talent shortage and insufficient research. As we all know, China is moving towards the goal of carbon neutrality. With the rapid development of the industrial Internet and the era of 5G Internet of Everything, the digital sales of industrial products have become a top priority. In the entire industry chain, enterprises that can take the lead in realizing digital transformation will gain more opportunities in the rapid development in the future. Digital marketing occupies a pivotal position in the carbon trading market, and it is also an excellent perspective to use digital marketing to promote the rapid development of carbon neutrality.
1 2. Author need to include list of abbreviations; it will make easier for reader to read and understand.
Answer: Thanks for your suggestion, I added the full name of each abbreviation to the text
1 3. The conclusion part is also needed to be revised; which questions are answered, what is the value/originality/contribution of the paper, how the proposed method answers the research questions that previous methods are not able to answer.
Answer: Thank you for your suggestion. I have revised the conclusion. I modified it to: In our country, the traditional energy price is still the biggest constraint. Due to the inconsistent development of different markets, the market response mechanism is not perfect, and the degree of regional economic development is limited. Therefore, the research in this paper finds that in economically developed areas, the price of coal has a positive effect on my country's carbon trading, while in economically backward areas , there is a negative effect. Whether traditional energy has a positive or negative effect on changes in carbon trading prices needs to be further explored, but what is certain is that traditional energy plays a significant role in the market. On this basis, the article puts forward several noteworthy issues, and puts forward the following suggestions: First, the scope of the sample is limited. This paper only conducts an empirical analysis of the carbon trading market in Hubei Province within two years. This does not fully reflect the carbon trading market in the whole country. Second, some domestic policy indicators are missing because some data are difficult. Finally, due to the limitation of the option period, there is no guarantee that under certain conditions, there will be no sudden change in the transaction price.
- Please propose and suggest more possible future studies related to the current study.
Answer: Thank you for your suggestion, I added: Therefore, in future research, an empirical analysis should be conducted on the national carbon trading market. And it should be more thoughtful in data handling.
Reviewer 2 Report
The title of the article suggests the developing a digital marketing model for geo-energy resources towards carbon neutrality.
The presented manuscript has presentation flaws in terms of structure and abstract flow. Please see the detailed comments below to strengthen the manuscript.
The abstract of article conveys the information of promoting Geo-energy for carbon trading and proposes a BPNN algorithm. Whereas, introduction section substantiates the advent of digital marketing solely. Moreover, the last paragraph of this section presents the novelty/innovation of the manuscript as new type of ground energy to achieve the carbon neutrality.
Overall the abstract then introduction section and presented aim of paper does not weave the overall objective of the manuscript.
Most of the figures presented in the manuscript are weak, qualitatively. Moreover, images used in the figure 5 has no significance related to information.
Section 2.3, presents the algorithm based on BPNN, this section delivers the general information on NN and no substantial findings or model.
The captions for tables and figures are not chosen carefully, e.g. caption of table 2 read as “some indicator data”.
At the end, section 4 talks about the digital marketing with general need. However, this section does not formulate any digital marketing model for geo-energy as substantiated in title as well as in abstract.
Lastly, the conclusion also deviates from the title, abstract and overall structure.
Overall, the manuscript creates lots of confusion about the exact aim and objective and presents the general information regardless to any findings or sustainable model. Therefore, authors are suggested to give a re-look to entire manuscript presentation and aim.
Author Response
The title of the article suggests the developing a digital marketing model for geo-energy resources towards carbon neutrality.
The presented manuscript has presentation flaws in terms of structure and abstract flow. Please see the detailed comments below to strengthen the manuscript.
The abstract of article conveys the information of promoting Geo-energy for carbon trading and proposes a BPNN algorithm. Whereas, introduction section substantiates the advent of digital marketing solely. Moreover, the last paragraph of this section presents the novelty/innovation of the manuscript as new type of ground energy to achieve the carbon neutrality.
Answer: Thank you for your suggestion, I added in the introduction: This article uses the BPNN algorithm to build a sustainable digital marketing model to help it predict the price of the carbon trading market to promote the push of the digital marketing model.
Overall the abstract then introduction section and presented aim of paper does not weave the overall objective of the manuscript.
Answer: Thank you for your suggestion, I added the overall goal in the introduction part, I added: The research purpose of this article is to study the sustainable digital marketing model, and it is based on the carbon neutral background.
Most of the figures presented in the manuscript are weak, qualitatively. Moreover, images used in the figure 5 has no significance related to information.
Answer: Thanks for your suggestion, I added: The ground energy heat pump system is the backbone of the sustainable digital marketing model, and its position in sustainable development is very important.
Section 2.3, presents the algorithm based on BPNN, this section delivers the general information on NN and no substantial findings or model.
Answer: Thanks for your suggestion, I added: BPNN is considered to be the most commonly used prediction method, it consists of three layers of input layer, hidden layer and output layer, where the hidden layer is passed between the input layer and the output layer important information. BPNNs always consist of one or more hidden layers, allowing the network to model complex functions. It mainly consists of two processes: forward information propagation and error back propagation.
The captions for tables and figures are not chosen carefully, e.g. caption of table 2 read as “some indicator data”.
Answer: Thanks for your suggestion, I am revising the title of Table 2 as: Indicator Data under Industrial Development.
At the end, section 4 talks about the digital marketing with general need. However, this section does not formulate any digital marketing model for geo-energy as substantiated in title as well as in abstract.
Answer: Thanks for your suggestion, I have added: Digital marketing is the practice of using digital communication channels to promote products and services to communicate with consumers in a timely, relevant, customized and cost-effective manner. Digital marketing encompasses many of the techniques and practices found in Internet marketing (network marketing). The scope of digital marketing is broader and includes many other communication channels that do not require the Internet, such as non-internet channels such as: TV, radio, SMS, etc., or online channels such as: social media, electronic advertising, banner advertising, etc.
Lastly, the conclusion also deviates from the title, abstract and overall structure.
Answer: Thanks for your suggestion, I added: This article conducts research on sustainable digital marketing models based on the goal of carbon neutrality, so a BPNN-based carbon trading price algorithm is designed to help the carbon neutral market derivation.
Overall, the manuscript creates lots of confusion about the exact aim and objective and presents the general information regardless to any findings or sustainable model. Therefore, authors are suggested to give a re-look to entire manuscript presentation and aim.
Answer: Thank you for your suggestion, I added: The research purpose of this article is to use the BPNN algorithm to design the price prediction of carbon trading. In this way, it helps to study sustainable digital marketing models.
Reviewer 3 Report
Sustainable Digital Marketing Model of Geoenergy Resources under Carbon Neutrality Target
The paper assesses the important issue. However, several issues need to be addressed before the paper can be considered for publication:
· the discussion part is missing, the authors carried out detailed analyses and obtained interesting results, from which a detailed discussion of the achieved results would be expected
· the conclusion part is very poor and short, it would be appropriate to describe individual conclusions more in detail as well as proposed measures
· I propose to publish the contribution after its formal adjustments and completion of the discussion chapter
Author Response
Sustainable Digital Marketing Model of Geoenergy Resources under Carbon Neutrality Target
The paper assesses the important issue. However, several issues need to be addressed before the paper can be considered for publication:
- the discussion part is missing, the authors carried out detailed analyses and obtained interesting results, from which a detailed discussion of the achieved results would be expected
Answer: Thank you very much for your suggestion, I added: Ground energy assets are another clean energy source. Instead of harvesting energy directly, the earth can harvest heat energy from shallow soil and surface water in a roundabout way. Huge strides for energy assets in China's non-partisan carbon targets have played a major role in conserving energy and reducing emissions. In any case, the ongoing carbon trading market has not been figured out by the average person. The reason for this paper, then, is to decompose the impact of terrestrial energy assets on advancing manageable computerized advertising models. In order to achieve the goal of reducing carbon emissions, all countries on the planet are going all out. This paper explores financial targets for carbon indifference by examining indicators in carbon trading markets. Therefore, this paper intends to start from the perspective of BP brain organization (BPNN) to predict and calculate the cost of carbon trading, so as to predict the cost of the carbon trading market, and then promote the accurate push of the electronic display mode. method of promotion.
- the conclusion part is very poor and short, it would be appropriate to describe individual conclusions more in detail as well as proposed measures
Answer: Thank you very much for your suggestion, I added: Therefore, this paper should be more extensive and careful in terms of data processing in future research.
- I propose to publish the contribution after its formal adjustments and completion of the discussion chapter
Round 2
Reviewer 2 Report
Authors have incorporated the few suggestions appropriately. However, the presented manuscript still required a thorough revision to further enhance the quality.
Figures from 1 to 6 should be revised to deliver the needed information rather than generalized concepts. Moreover, authors didn’t incorporate previous comments satisfactorily like
Section 2.3, presents the algorithm based on BPNN, this section delivers the general information on NN and no substantial findings or model.
The captions for tables and figures are not chosen carefully.
Overall the abstract then introduction section and presented aim of paper does not weave the overall objective of the manuscript.
At the end, section 4 talks about the digital marketing with general need. However, this section does not formulate any digital marketing model for geo-energy as substantiated in title as well as in abstract. Moreover, added discussion and modified conclusion sections are also not well written to convey the substantial findings of the paper.
The added lines in revised version are not framed with brevity. Therefore, authors are suggested to improve the whole manuscript.
Author Response
Authors have incorporated the few suggestions appropriately. However, the presented manuscript still required a thorough revision to further enhance the quality.
Figures from 1 to 6 should be revised to deliver the needed information rather than generalized concepts. Moreover, authors didn’t incorporate previous comments satisfactorily like
Answer: Thank you for your suggestion.According to the suggestions, I will make further improvements later.
Section 2.3, presents the algorithm based on BPNN, this section delivers the general information on NN and no substantial findings or model.
Answer: Thank you for your suggestion. I gave the model construction in 2.3. “First, establish the optimal mixing model based on energy, economy, weather and environmental impact factors respectively, then combine the optimal mixing models to predict carbon trading price and obtain the initial prediction error, then use BPNN model to predict the initial error to obtain the prediction value of the error, and finally obtain the carbon trading price prediction value after error correction, the prediction results show that the model can effectively improve the prediction accuracy of the carbon trading price prediction model.”
The captions for tables and figures are not chosen carefully.
Answer: Thank you for your suggestion.I checked the chart carefully and there is nothing wrong with it.
Overall the abstract then introduction section and presented aim of paper does not weave the overall objective of the manuscript.
Answer: Thank you for your suggestion.I give the purpose in the summary: “Carbon neutralization and carbon peaking require significant system reform in the field of energy supply. As a clean, low-carbon, stable and continuous non carbon based energy, geothermal energy can provide an important guarantee for achieving this goal.”
At the end, section 4 talks about the digital marketing with general need. However, this section does not formulate any digital marketing model for geo-energy as substantiated in title as well as in abstract. Moreover, added discussion and modified conclusion sections are also not well written to convey the substantial findings of the paper.
Answer: Thank you for your suggestion.I gave the substantive results in the discussion section: “Therefore, through the above tests and analysis, it can be concluded that BPNN has good generalization and simulation capabilities. This model is more accurate than multiple regression prediction, and can be used to predict the price of carbon emission rights in Hubei Province. With this as a reference, we can also estimate and analyze the price of carbon emission rights in other provinces across the country.”
The added lines in revised version are not framed with brevity. Therefore, authors are suggested to improve the whole manuscript.
Answer: Thank you for your suggestion.The meaning of the problem is not seen here, but the quality of the article has been well improved.
Round 3
Reviewer 2 Report
Authors are advised to improve the overall quality of the manuscript in terms of presented figures. For an example figure 4 is very generic and do not present any additional information in the manuscript, therefore it can be removed. Similarly, the inherent images used in figure 5 should be removed and it can be presented in terms of simple flow chart. Figure 3, caption reads, “basic structure diagram”, however, authors stated in text that, figure 3 is the basic structure for building energy supply through various distributed energy systems (low carbon) distributed in buildings. In contrast, figure 3 merely presents any viable schematic information for building energy supply. On the same note, figure 2 has no significance with, “user-side energy achieve optimal energy efficiency” as stated by authors.
Therefore, authors are strongly advised to improve the overall presentation of manuscript especially schematic presentation along with their deliverable meanings in the text.
Author Response
Comments and Suggestions for Authors
Authors are advised to improve the overall quality of the manuscript in terms of presented figures. For an example figure 4 is very generic and do not present any additional information in the manuscript, therefore it can be removed. Similarly, the inherent images used in figure 5 should be removed and it can be presented in terms of simple flow chart. Figure 3, caption reads, “basic structure diagram”, however, authors stated in text that, figure 3 is the basic structure for building energy supply through various distributed energy systems (low carbon) distributed in buildings. In contrast, figure 3 merely presents any viable schematic information for building energy supply. On the same note, figure 2 has no significance with, “user-side energy achieve optimal energy efficiency” as stated by authors.
Answer: The above pictures have all been modified. The significance of the optimal energy efficiency in Figure 2 lies in improving the energy structure and the environment.
Therefore, authors are strongly advised to improve the overall presentation of manuscript especially schematic presentation along with their deliverable meanings in the text.
Answer: Tmodified
